# Numerical Study on the Aeroacoustic Performance of Different Diversion Strategies in the Pantograph Area of High-Speed Trains at 400 km/h

Hongkang Liu [1,2,3], Siqi Zhou [1,2,3], Rongrong Chen [1,2,3], Zhuolun Li [1,2,3], Shishang Zhang [1,2,3] and Yatian Zhao [1,2,3,*]

1  Key Laboratory of Traffic Safety on Track of Ministry of Education, School of Traffic & Transportation Engineering, Central South University, Changsha 410075, China
2  Joint International Research Laboratory of Key Technology for Rail Traffic Safety, Central South University, Changsha 410075, China
3  National & Local Joint Engineering Research Centre of Safety Technology for Rail Vehicle, Central South University, Changsha 410075, China
*  Correspondence: yatianzhao@126.com

**Abstract:** The speed increase in high-speed trains is a critical procedure in the promotion of high-speed railway technology. As an indispensable and complex structure of high-speed trains, the pantograph's aerodynamic drag and noise is a significant limitation in the speed increase process of high-speed trains. In the present study, the hybrid method of large eddy simulation (LES) and Ffowcs Williams-Hawkings (FW-H) acoustic analogy is applied to analyze the aerodynamic and aeroacoustic performances of pantograph installed in different ways, i.e., sinking platform and fairing. The results of simulation show that the application of pantograph fairing can reduce the aerodynamic drag greatly. In addition, compared with the pantographs installed alone on the train roof, the installation of the sinking platform brings about 2 dBA reduction in sound pressure level (SPL). Meanwhile, the utilization of the pantograph fairing mainly decreases the noise in the frequency band above 1000 Hz and the largest SPL reduction is up to 3 dBA among the monitoring points. Further analysis shows that the influence of different diversion strategies on the spectral characteristics actually attenuates the dominant frequency of the panhead. In the horizontal plane, the noise directivity of the pantograph installed with a fairing is similar to the pantograph installed alone on the train roof.

**Keywords:** aerodynamic noise; aerodynamic drag; pantograph sinking platform; pantograph fairing; high-speed train

## 1. Introduction

High-speed trains (HSTs) stand out in the competition of many means of transportation by virtue of their high efficiency, safety, comfort, and environmental friendliness. With the deepening of theoretical research and the gradual maturity of technical development, China's 350 km/h HSTs have been put into operation, and the development of higher-speed trains has become the pursuit goal of researchers in related industries at home and abroad [1,2]. However, there are significant aerodynamic problems in the process of speed increase in HSTs. Aerodynamic noise is proportional to the sixth power of the train running speed. When the train running speed exceeds 300 km/h, aerodynamic noise will be the main noise source, and the proportion of aerodynamic drag will also be significantly increased. Strong annoying noise not only affects the comfort of passengers, but also interferes with the normal life of surrounding residents [3,4]. Therefore, it is essential to control the aerodynamic noise.

Talotte [5] considered that the critical aspect of any acoustical study is to ascertain noise sources, and revealed that the predominant aerodynamic noise sources of HSTs are

the pantograph, the sinking platform of the pantograph, the inter-coach spacing, the bogie, the noise of the leading and rear car, the surfaces, louvres, and ventilators. The noise of HSTs was divided into two categories. The first was the noise generated by the flow flowing through various structural components, such as the vortex shedding sound generated by the flow flowing through the pantograph rod, and the cavity noise from the pantograph recess. The second was the turbulence-generated noise, which was mainly generated at the turbulent boundary layer or its separation on the surface of HSTs. Kitagawa and Nagakura [6] conducted a noise spectrum analysis of the data collected with a microphone array and concluded that after the train's surfaces were smoothed, the pantograph and the bottom of the train became the main noise sources. Moreover, in 2006, Nagakura [7] found that the pantograph is one of the strongest noise sources, and concluded that the bottom of the head car is the leading noise source, followed by the cab door and the windshield through wind tunnel experiments. The author hypothesized that these noises can be attenuated by sound barriers, thus the gap at the top of the train becomes the most notable noise source. King and Pfizenmaier [8] measured the noise of cylinders with various cross-sections in a wind tunnel with a maximum speed of 67 m/s. The results showed that cylinders with elliptical cross-sections or knurled surfaces were less noisy and were expected to be used in pantographs for HSTs.

As the dominant noise source of HSTs and whose noise cannot be isolated by sound barriers, numerical simulations of pantographs have been carried out by many researchers. This is due to the benefits of numerical simulation, which shows its advantage of short period and low cost. In addition, it can be related to the Lighthill analogy or the resolution of the Euler equation based on sound propagation, making it one of the most widely used research methods [9–12]. Yu et al. [13] conducted a hybrid method of non-linear acoustics solver (NLAS) and Ffowcs Williams-Hawkings (FW-H) acoustic analogy to predict the noise of the pantograph system. In addition, the noise reduction effect of four types of pantograph covers was compared. The authors found that the aerodynamic noise of the pantograph radiates outward in the far field as spherical waves, and only the cover consisting of baffles on both sides was effective for noise reduction. Zhang et al. [14] utilized large-eddy simulation (LES) and FW-H equation to explore the flow characteristics near the pantograph and the far-field radiated noise, and the results showed that the primary noise sources of the pantograph are the panhead, the base frame, and knuckle. Tan et al. [15] considered the Faiveley CX-PG pantograph as the research object and simulated the turbulent flow characteristics and far-field radiated noise around it by LES and FW-H equation to explore the connection between the vortex structure near the pantograph and the acoustic performance. The authors found that the sound source intensity distribution of the pantograph is closely associated with the position of vortex shedding, and whether the pantograph is immersed in the wake flow of the vortex. Yao et al. [16] applied LES and the acoustic finite element method (FEM) to analyze the acoustic spectrum characteristics and aerodynamic noise distribution in the near-field and far-field regions of pantographs with different installation bases: Flush and sunken. It was found that the sunken pantograph installation platform has better aerodynamic noise performance. Kim et al. [17] studied the influence of cavity or sinking platform on pantograph aerodynamic noise, and compared the aerodynamic performance of the pantographs between the pantpgraphs installed in recesses and on the roof alone. The flow field of the pantograph was simulated by improved delayed detached-eddy simulation (IDDES) and the far-field noise was computed by the FW-H analogy. In addition, it was confirmed that the cavity could reduce the pressure fluctuating on the surface of the pantograph, which in turn reduced its noise. Zhao et al. [18] predicted the near-field and far-field noise of the pantograph by LES, acoustic perturbation equation (APE), and FW-H equation. They divided the vortex structure behind the pantograph into the panhead area, the middle area, and the groove area. Moreover, they stated that only the dipole source was regarded as the source of far-field radiation noise since the intensity of the quadrupole source was substantially less. The authors of [19] studied the aerodynamic performance of two new pantograph forms by IDDES and FEM, and found that

the pantograph with smoother insulators had no prominent effect in terms of noise, while the pantograph with smoother base frame had lower drag and noise. Shaltout et al. [20] compared different cross-sectional panheads and showed that at 250 km/h, the elliptical cross-sectional panhead could reduce the sound pressure level (SPL) of the pantograph by over 20%.

The above numerical studies for pantograph noise reduction can be divided into changing the installation type or shape of pantograph [21–23]. Moreover, the variation of aerodynamic drag under the new type of pantograph is rarely considered at the same time. Furthermore, changing the rod type of the pantograph may have an impact on its receiving current. Studies have shown that the pantograph cover becomes a feasible solution to achieve the drag reduction requirements at higher speeds. However, its effect on aerodynamic noise is still unclear and remains to be specified, and this is the purpose of this paper.

## 2. Numerical Simulation Methodology

The numerical methods of aerodynamic noise are categorized into direct noise simulation and hybrid method. Due to the large difference in scale and energy between the acoustic and flow field, a discrete format with high order accuracy, low dissipation, and low dispersion is needed to improve the solution accuracy, which greatly reduces the feasibility of the direct method. Particularly for HSTs and their components with complex geometry, the hybrid method is usually used to predict the aerodynamic noise. The hybrid solution divides the numerical simulation of aeroacoustic performance into near-field and far-field solutions, i.e., sound source simulation and sound propagation solution.

In the present study, CFD simulations are proceeded by ANSYS Fluent software. The LES turbulence model is used to solve the flow-field structure near the pantograph first, and then the FW-H analogy equation is applied to predict the far-field noise of the pantograph area.

### 2.1. LES Method

Turbulent flows are characterized by disordered, eddying fluid motions over a variety of length scales. The size of largest eddies is almost comparable to the characteristic length of the mean flow, and the dissipation of turbulence kinetic energy can be ascribed to the smallest scales. In LES, the basic idea is to establish a mathematical filter function to separate and filter large-scale eddies and small-scale eddies in turbulent flows, where the turbulent motion of large-scale eddies is resolved directly by the Navier-Stokes equation (N-S equation), and small-scale eddies are modeled by a closed model to show the dissipation relation for large-scale quantities [24].

The Filter function equation is as follows:

$$\overline{\phi} = \frac{1}{V} \int_D \phi dx \tag{1}$$

where $\phi$ is the transient turbulent fluctuation volume and $V$ denotes the occupied geometric space of the governing volume.

The N-S equation for a compressible fluid filtered by a function takes the form of:

$$\frac{\partial \rho}{\partial t} + \frac{\partial \rho \overline{u}_i}{\partial x_i} = 0 \tag{2}$$

$$\frac{\partial}{\partial t}(\rho \overline{u}_i) + \frac{\partial}{\partial x_j}(\rho \overline{u}_i \overline{u}_j) = \frac{\partial}{\partial x_j}\left(\mu \frac{\partial \overline{u}_i}{\partial x_j}\right) - \frac{\partial \overline{p}}{\partial x_i} - \frac{\partial \tau_{ij}}{\partial x_j} \tag{3}$$

where $\tau_{ij}$ is the subgrid-scale stresses term and $\tau_{ij}$ is defined as $\tau_{ij} = \rho \overline{u_i u_j} - \rho \overline{u}_i \cdot \overline{u}_j$. Moreover, $u$ is the flow velocity, $\rho$ is the density, and $\overline{p}$ is the filtering pressure.



Using the filtering operation, the subgrid-scale stresses are introduced and require modeling. To close the filtered N-S Equations (2) and (3), the subgrid-scale (SGS) model based on Boussinesq hypothesis is employed widely. The corresponding SGS model is:

$$\tau_{ij} - \frac{1}{3}\tau_{kk}\delta_{ij} = -2\mu_t\overline{S}_{ij} \tag{4}$$

where $\mu_t$ is the subgrid-scale turbulent viscosity, $\tau_{kk}$ is the isotropic part of the subgrid-scale stresses, and $\overline{S_{ij}}$ is the rate-of-strain tensor for the resolved scale defined by Equation (5):

$$\overline{S}_{ij} = \frac{1}{2}\left(\frac{\partial\overline{u}_i}{\partial x_j} + \frac{\partial\overline{u}_j}{\partial x_i}\right) \tag{5}$$

The Smagorinsky-Lilly model is chosen here and $\mu_t$ is defined as $\mu_t = \rho L_s^2\left|\sqrt{2\overline{S}_{ij}\overline{S}_{ij}}\right|$, where $L_s$ represents the mixing length for subgrid scales.

### 2.2. Ffowcs Williams-Hawkings Acoustics Analogy

In 1969, Ffowcs Williams-Hawkings applied the generalized function method to extend Curle's result to the sound generation problem of arbitrary moving surfaces in an unbounded fluid. The FW-H equation is essentially an inhomogeneous wave equation that can be derived by manipulating the continuity equation and N-S equation [25,26]. It can be written as:

$$\frac{1}{a_0^2}\frac{\partial^2 p'}{\partial t^2} - \nabla^2 p' = \frac{\partial^2}{\partial x_i\partial x_j}\left\{T_{ij}H(f)\right\} - \frac{\partial}{\partial x_i}\left\{[P_{ij}n_j + \rho u_i(u_n - v_n)]\delta(f)\right\}$$
$$+ \frac{\partial}{\partial t}\left\{[\rho_0 u_n + \rho(u_n - v_n)]\delta(f)\right\} \tag{6}$$

where $u_i$ and $v_i$ represent the flow velocity and surface velocity component in the $x_i$ direction, $u_n$ and $v_n$ represent the flow velocity and surface velocity normal to the surface, $\delta(f)$ is the Dirac delta function, and $H(f)$ is the Heaviside function. In addition, $p'$ denotes the sound pressure at the far-field ($p' = p - p_0$), $a_0$ is the far-field sound speed, and $T_{ij}$ is the Lighthill stress tensor, defined as:

$$T_{ij} = \rho u_i u_j + P_{ij} - a_0^2(\rho - \rho_0)\delta_{ij} \tag{7}$$

where $P_{ij}$ is the compressive stress tensor. Equation (6) contains the effects of all the sound sources in the flow field: The first term on the right reflects the quadrupole sound caused by the turbulent flow, the second term is the dipole sound caused by the solid surface force, and the third term is the monopole sound caused by the solid boundary motion.

In the present study, the pantograph surfaces are set as integration surfaces and can be regarded as arbitrary rigid bodies; namely, the fluctuating volume quantity turns to zero. Therefore, there is no need to consider the monopole sources. Meanwhile, the Mach number (*M*) is set as 0.33 and the corresponding incoming flow velocity is 400 km/h. Moreover, the authors of [14] noted that the ratio of quadrupole source intensity to the dipole source intensity in flow field is proportional to $M^2$. The noise intensity induced by quadrupole sources is significantly less than the dipole source, and thus the quadrupole sources can be ignored.

### 2.3. Validation of the Methods

Herein, we considered the lack of aerodynamic noise experiment data for real pantographs and the inevitable simplification of pantographs for numerical simulations. For the purpose of verifying the reliability of the computational method, a classical finite length cylindrical flow is selected for numerical simulation in this paper. The rod-airfoil experiment was carried out by Jacob et al. [27] and was introduced as a validation case in a previous study on pantograph noise [13]. The diameter of the rod (D) obtained for

the experiment is 10 mm, and the spanwise length is 30 D. The incoming flow velocity is 72 m/s and the free flow turbulence intensity is set as 0.8%. The Reynolds number based on the diameter is about $4.8 \times 10^4$, based on rod diameter. Figure 1 displays the computational domain for the validation. The size of 40 D in the stream-wise direction and 20 D in the corresponding normal direction is employed. In addition, considering the computational resources due to the large amount of mesh brought about by the experimental spanwise length, here the spanwise length is chosen as πD according to references [13,28–30].

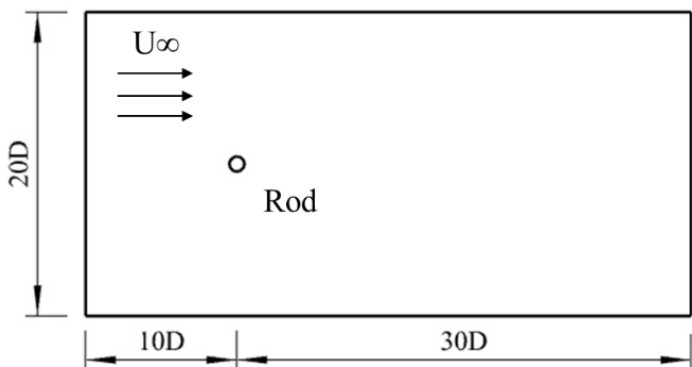

**Figure 1.** Schematic diagram of the computational domain for the validation.

The computational mesh is shown in Figure 2. Two refinement zones are used to refine the volume in the wake flow region of the rod. To make y+ values meet the requirement and y+ < 1 in this paper; the first layer thickness is 0.004 mm and the growth ratio of 1.1 is given in the wall-normal direction. The radial grid near the wall is generated according to reference [13,28]. The total number of grid cells is about 1 million. Moreover, the size of grid cells in the spanwise direction is about 1 mm and the spanwise direction is uniformly distributed with about 33 grid cells. In a previous study, the authors of [31] found that the two-point pressure correlation length is about 3 D, indicating that the spanwise space in this study is physically reasonable.

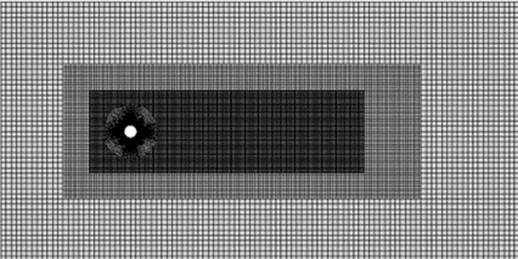

**Figure 2.** Schematic diagram of the computational mesh for the validation.

To be consistent with the study by Yu et al. [13], the velocity inlet and pressure outlet boundary conditions are prescribed. The incoming free flow velocity is 72 m/s, which is consistent with the experiment. Of note, both the velocity inlet and pressure far-field conditions are compared here and the results show few discrepancies. The latter is used in the following simulations of the pantograph. In addition, the turbulence intensity is set to 0.8%. For the spanwise and normal directions, symmetry boundary conditions are used. Contrarily to the periodic condition, the symmetry condition only makes the velocity components in the z-direction disappear and does not correlate the aerodynamic flow field variables in the existing planes. In addition, the two-point spanwise pressure correlation on the separate rod was in good agreement with the experimental results, which is very important for acoustic calculations and cannot be acquired with the periodic condition. Therefore, the symmetry condition should be a better choice here [32].

Table 1 presents the comparison of the drag coefficients. Of note, the average drag coefficient and its fluctuating values are in good agreement with those reference experimental results, indicating the rationality of the computation method in this study.

**Table 1.** Computed forces in comparison with the available experimental data.

| Data Sources | Method | Re | $C_D$ | $C'_{Drms}$ |
|---|---|---|---|---|
| Present study | LES | $4.8 \times 10^4$ | 1.0 | 0.081 |
| Cantwell and Coles [33] | EXP | / | [1.0, 1.35] | / |
| Gerrard [34] | EXP | $4.0 \times 10^4$ | / | [0.08, 0.1] |
| Boudet et al. [35] | LES | $4.6 \times 10^4$ | 1.02 | 0.076 |

Herein, we considered the effect of spanwise correlation, according to reference [35]. At the sound monitoring point, which is located at a distance of 185 D in the normal direction of the flow, the far-field SPL versus the Strouhal number is displayed in Figure 3. It is shown that the base level of the spectrum is captured. Specifically, the acoustic peak of nearly 90 dB appears at St = 0.197, indicating the shedding frequency of Karman vortex. Meanwhile, the second and third modes can also be accurately predicted by the present numerical simulations. This performance indicates that the current computational methodology is capable of qualitatively predicting aerodynamic noise.

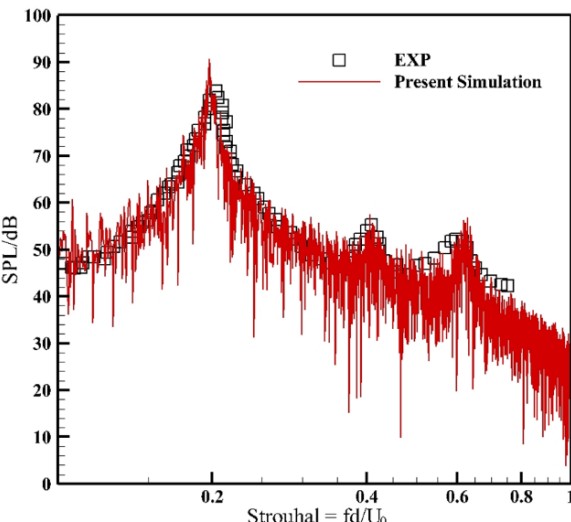

**Figure 3.** Acoustic pressure results compared with the experimental data [27] at (0.105, 0.5 $\pi$D, 185 D).

## 3. Computational Configurations

### 3.1. Computational Models

The pantograph is a current receiving device, in which the train receives current from the contact wire of the catenary. It is installed on the roof of the train, and when the pantograph is raised, the current can be introduced into the train carriage through the contact between the components and the contact wire. The pantograph used in this study is a single-arm pantograph, which consists of upper arm bar, lower arm bar, balance bar, bow head, base frame, insulator, camera, and other parts as shown in Figure 4. In addition, the height of H is from the bottom of the train to the top of the pantograph, which is defined as the characteristic length in the numerical simulation, as depicted in Figure 5. When a sinking platform occurs, the height of the pantograph remains the same. The sinking height is 0.15 m, which is relatively small compared to the characteristic length H.

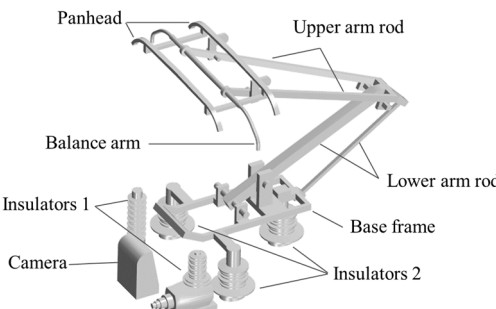

**Figure 4.** Computational model of pantograph.

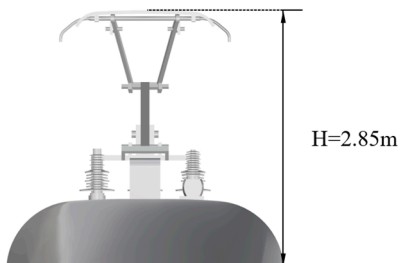

**Figure 5.** Side view of the computational model.

### 3.2. Computational Domain and Numerical Setup

The computational model used in this paper is the pantograph placed on the train roof model with three cars to ensure the reasonability of the inflow and reduce the mesh amount. In addition, the computational domain is sufficiently large to eliminate the effects of the surrounding boundaries in the numerical simulation [36]. The origin of the used coordinate is at the bottom center of the train roof model and the inlet flow is due to the negative direction of *x*-axis. Moreover, the size of the computational domain is $60 \times 20 \times 10$ H, where H = 2.85 m, as depicted in Figure 5. To ensure that the inflow develops fully before reaching the head car, the upstream length is 10 H. Similarly, the downstream length is longer than 20 H to ensure that the vortex shedding in the wake flow is well-developed. The computational domain is shown in Figure 6. This study mainly focuses on the effects of pantograph diversion strategies on train aerodynamic and aeroacoustic performance. Three configurations designed here, including the pantograph installed on the train roof alone, on the train roof sinking platform, and the pantograph fairing adopted on the train roof are described in Figure 7 and Table 2.

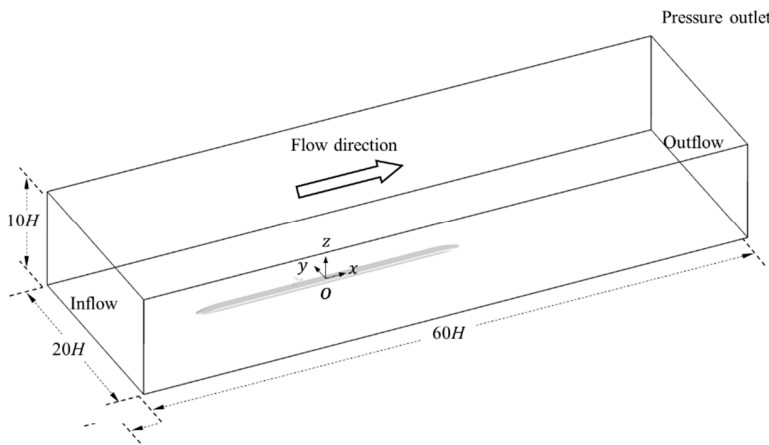

**Figure 6.** Sketch of the computational domain.

Figure 7b illustrates the details of the sinking platform. For the sinking platform, its sinking height is 0.15 m, the width of front edge is 2.1 m, and the width of back edge is

0.9 m. The length of the sinking platform is 3.5 m in x direction. For the fairing, the specific parameters are clearly shown in Figure 7c as 8.3 and 2.3 m, respectively, and the largest height from the roof is 0.6 m, which is located at the rear edge of the sinking platform. To reduce the self-resistance, it remains streamlined at the front and back surface.

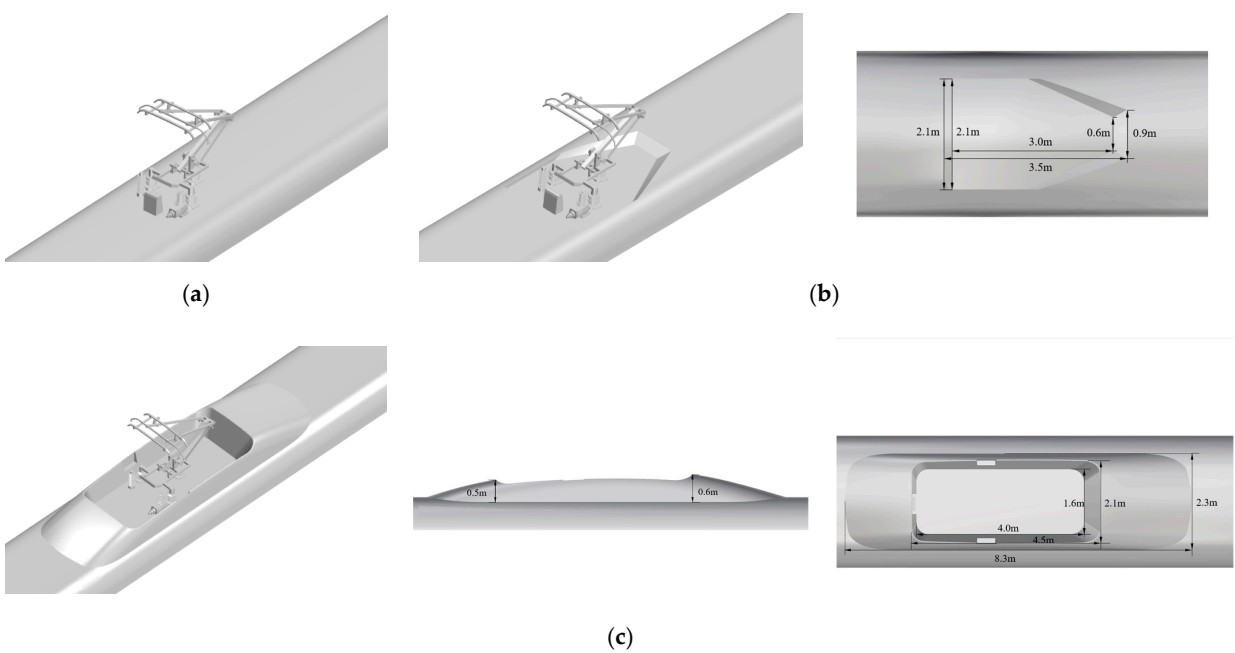

(a)

(b)

(c)

**Figure 7.** Sketch of the computational models. (**a**) Case 1; (**b**) case 2; (**c**) case 3.

**Table 2.** Three configurations and their so-called names.

| / | Configurations |
|---|---|
| Case 1 | Pantograph installed on the train roof alone |
| Case 2 | Pantograph installed on the sinking platform |
| Case 3 | Pantograph faring installed on the train roof |

To preclude the influence of grid, a grid independence study is carried out. Three grid resolutions including coarse, medium, and fine grids are set and the total drag coefficient of case 1 is compared, as shown in Table 3. It can be found that the calculated drag coefficient of case 1 shows a larger difference when the coarse and medium grids are adopted. In addition, the relative error in the calculated results between the medium and fine grid models is less than 2%. This indicates that the flow field computation results are reliable when the computational domain reaches the medium grid scale. The medium computational mesh is depicted in Figure 8. It is discretized by the poly-hexcore unstructured grid. The grid size of the pantograph surface is less than 8 mm to guarantee the prediction accuracy of the pantograph aerodynamic noise calculation. In addition, the boundary layer grids are generated near the pantograph surface, obtaining the first layer thickness of 0.12 mm which is normal to the wall, to ensure y+ ≈ 1 with a growth ratio of 1.2. To capture the flow details, the refinement zone is carried out on the leeward side of the pantograph and the tail flow area of the roof model.

The pressure far-field condition and the pressure outlet condition are prescribed for the inlet and outlet boundary here, respectively. Both side planes of the computational model and the top planes are set as symmetry boundaries. To simulate the ground effects, the ground is set to be the moving wall condition with an incoming velocity of 400 km/h. The physical time step of large eddy simulation is considered as $5 \times 10^{-5}$ s, and the corresponding aerodynamic noise for analysis frequency is 10 kHz, according to the Nyquist sampling theorem [37]. Moreover, 25 sub-steps are iterated within each time step and a total

of 8000 time steps are calculated. The first 2500 time steps are calculated to ensure that the turbulent flow field is fully developed, and the remaining 5500 time steps are used to extract the noise source information.

**Table 3.** The validation of grid independence.

| Grids | Total Number of Grid Cells | Drag Coefficient | Relative Error |
|---|---|---|---|
| Coarse grid | 24 million | 0.5419 | 6.76% |
| Medium grid | 30 million | 0.5700 | 1.93% |
| Fine grid | 36 million | 0.5812 | / |

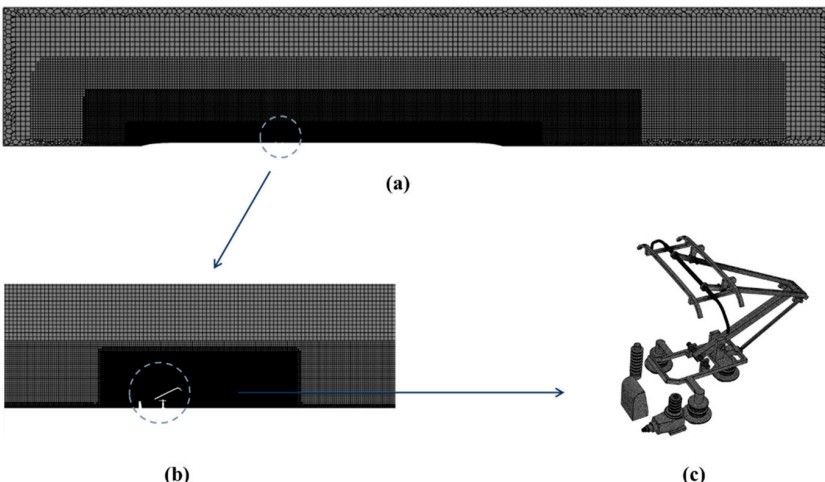

**Figure 8.** Sketch of the computational mesh. (**a**) Y-plane of the computational mesh; (**b**) enlarged view of the pantograph area; (**c**) enlarged view of the pantograph.

## 4. Results and Discussion

### 4.1. Near-Field Flow Field Characteristics

#### 4.1.1. Drag Coefficient of Different Pantograph Diversion Strategies

The following tables show the drag coefficients of the three calculated models, where Table 4 shows the drag coefficients in total and Table 5 shows the drag coefficients of each pantograph component. It can be found that the drag coefficients of the whole computational model decrease to different degrees when the sinking platform is installed on the train roof or after the pantograph fairing is installed. As shown in Table 4, compared with the pantograph exposed on the roof alone, the drag reduction rates of the above two cases are 2.93% and 5.42%, respectively. To obtain a better idea of how the total drag coefficients changed, it is necessary to analyze the drag of each pantograph part. As shown in Table 5, the drag coefficients of the base frame, insulator 1, and camera are dropped when the sinking platform is installed on the train roof, due to the reduction in their windward area. When the pantograph fairing is installed, most of the components are shielded by the fairing, especially the base frame, insulators, and camera; therefore, their drag coefficient decreases sharply. Although the drag coefficient of the fairing is comparable to the entire pantograph, as shown in Table 4, the total drag coefficient of the train model with three cars roof is greatly reduced after the installation of the fairing. As a consequence, the pantograph fairing could be a good alternative in terms of drag optimization for higher-speed train.

**Table 4.** Computed drag coefficients of three cases.

| | Head Car | Middle Car | Tail Car | Pantograph | Fairing | Total |
|---|---|---|---|---|---|---|
| Case 1 | 0.1022 | 0.0754 | 0.0762 | 0.3162 | / | 0.5700 |
| Case 2 | 0.1022 | 0.0833 | 0.0776 | 0.2902 | / | 0.5533 |
| Case 3 | 0.1008 | 0.0693 | 0.0829 | 0.1443 | 0.1418 | 0.5391 |

**Table 5.** Computed drag coefficients of each pantograph part.

| | Panhead | Balance Arm | Upper Arm Rod | Lower Arm Rod | Base Frame | Insulator 1 | Insulator 2 | Camera |
|---|---|---|---|---|---|---|---|---|
| Case 1 | 0.0709 | 0.0229 | 0.0287 | 0.0185 | 0.0590 | 0.0412 | 0.0132 | 0.0619 |
| Case 2 | 0.0695 | 0.0225 | 0.0292 | 0.0173 | 0.0528 | 0.0357 | 0.0149 | 0.0483 |
| Case 3 | 0.0726 | 0.0252 | 0.0286 | 0.0141 | 0.0073 | 0.0001 | −0.0037 | 0.0001 |

### 4.1.2. Flow Characteristics of Different Configurations

Figure 9 shows the normalized Q-criterion-based instantaneous iso-surface vorticity for different pantograph diversion strategies, colored by magnitude velocity. As seen in Figure 9, the incoming airflow directly impacts the pantograph surface and then separates, forming a clear separated vortex on the leeward side of the panhead, balance arm, and base frame. When the pantograph is installed on the train roof alone, the airflow first forms a series of small, flocculated vortices on the train surface while in front of the camera. The large vortex is formed on the pantograph surface with a large velocity at the beginning, and its velocity magnitude gradually decreases as the vortex moves backward away from the pantograph. When there is a sinking platform on the roof, the airflow is separated at the front edge of the sinking platform first, and a series of hairpin-shaped vortices are formed inside the platform. Then, the airflow that entered the sinking platform flows through the pantograph, and will separate again due to the interference of the pantograph, but with a different separated vortices shape when the pantograph is installed on the train roof alone. However, the airflow above the sinking platform is similar to case 1, and there are many fragmentized small-scale vortices. When the pantograph fairing is installed, the airflow first impacts the leading edge of the fairing, generating a small number of low-speed vortices, and then the disturbed incoming airflow continues to impact the pantograph surface, forming a series of hairpin-shaped large-scale vortices, which then impacts the rear of the fairing. These results are consistent with reference [38]. Compared with the former configurations, a large number of hairpin-shaped separated vortices are generated when the airflow enters the cavity due to the existence of the fairing. The high-speed airflow impacts the rear of the pantograph fairing, generating more separated vortices which are larger in scale, and elongate in the downstream.

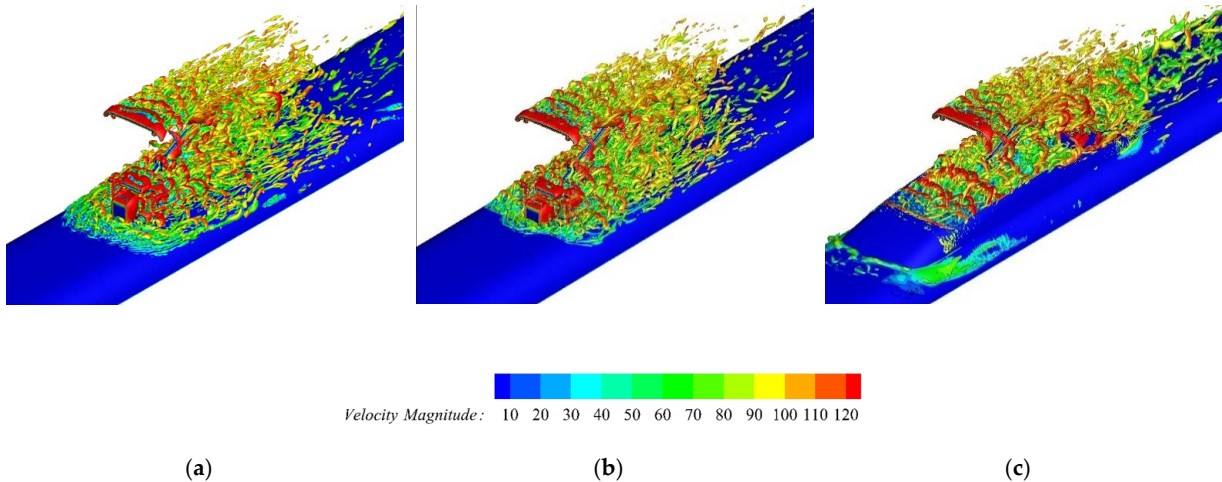

Velocity Magnitude :　10　20　30　40　50　60　70　80　90　100 110 120

(**a**)　　　　　　　　　　　　　　　　(**b**)　　　　　　　　　　　　　　　　(**c**)

**Figure 9.** Iso-surface of the instantaneous normalized Q-criterion. (**a**) Case 1; (**b**) case 2; (**c**) case 3.

### 4.2. Acoustic Characteristics of Different Configurations

#### 4.2.1. Aerodynamic Noise Source

This section analyzes the intensity characteristics of the aerodynamic noise dipole source using the root mean square of the time derivative of the surface fluctuating pressure.

After completing the instantaneous calculation, the time gradient p′ of the fluctuating pressure on the vehicle surface is calculated by Equation (8), and then the root mean square $p'_{rms}$ is computed by Equation (9) as follows:

$$p\prime = dp/dt \tag{8}$$

$$p\prime_{rms} = \sqrt{\frac{\int_{t_1}^{t_2} (p\prime)^2 dt}{t_2 - t_1}} \tag{9}$$

Figure 10 shows the $p'_{rms}$ distributions for the three pantograph installation methods. As seen in Figure 10, the pantograph sound source intensity is mainly distributed at the panhead, lower arm rod, base frame, etc. When the pantograph is installed on the train roof alone, the sound source intensity is also distributed at the balance arm. Combined with Figure 9 for analysis, the incoming airflow directly impacts the front edge of the panhead without interference, and vortex shedding occurs at the balance arm. When the sinking platform is installed on the train roof, the pressure fluctuation at the balance arm and the leeward side of the camera is weakened. When the pantograph fairing is not installed, the great sound source intensity also appears in the bottom area of the insulators. The installation of the fairing changes the position of vortex shedding when the airflow hits the pantograph. Consequently, the pressure fluctuation at the top of the insulator was stronger, and the strong pressure fluctuation area was also distributed at the front edge of the rear end of the fairing. In general, the pantograph pressure fluctuation is distributed in the area that is seriously affected by vortex shedding.

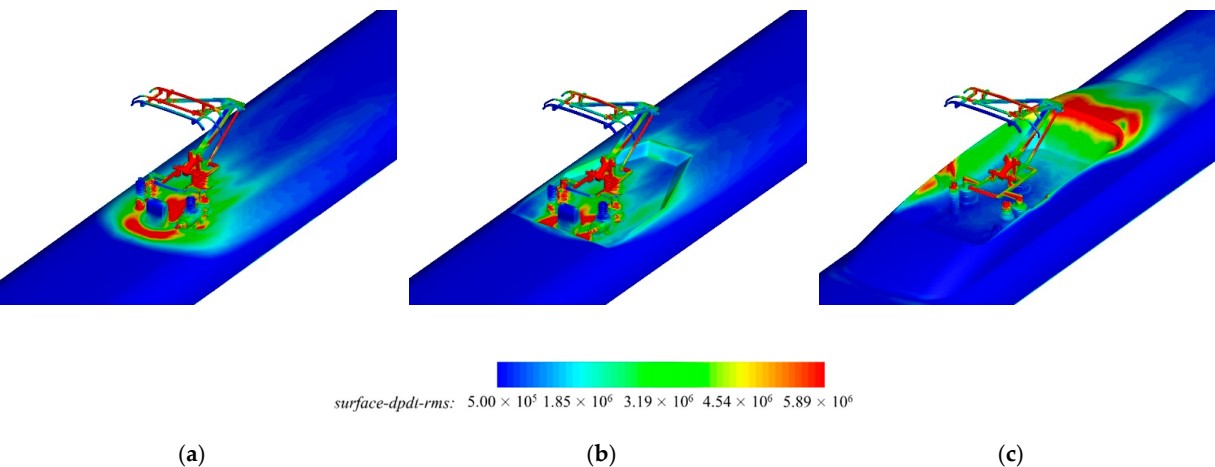

surface-dpdt-rms:   5.00 × 10⁵  1.85 × 10⁶  3.19 × 10⁶  4.54 × 10⁶  5.89 × 10⁶

(**a**)                                    (**b**)                                    (**c**)

**Figure 10.** Pressure fluctuation of the computational models. (**a**) Case 1; (**b**) case 2; (**c**) case 3.

4.2.2. Spectral Characteristics of SPL

This section studies the effect of different diversion strategies on the far-field radiated noise of pantographs. The standard observing points should be 25 m from the central line of rail track and 3.5 m above the surface of rail track. In view of the train roof model which is used in this study, the far-field acoustic measurement points are arranged at a height of 1.5 m from the ground and 25 m from the center line of rail track. As depicted in Figure 11, 20 monitoring points are equally spaced along the longitudinal direction of the train (x direction), each observer point is 5 m apart, and the origin is at the bottom center of train. Therefore, the range of monitoring points is [−35, 60] m. The pantograph is located near the 6th monitoring point along the x direction.

Figure 12 demonstrates the comparison curves of the SPL at monitoring points under different pantograph diversion strategies when the train model and pantograph are set as the sound sources. As shown in Figure 12, along the direction of train running, the aerodynamic noise SPL tends to increase first and then decrease, and the maximum SPL appears near the

pantograph area. Compared with the SPL when pantograph is installed on the train roof alone, it is clear that the SPL is reduced by about 2 dBA when the pantograph is installed on the sinking platform, indicating a better aeroacoustic performance. In addition, the maximum SPL of cases 1 and 2 occurs at the 6th monitoring point. When the pantograph is installed with the fairing, the location changes, which is the 4th monitoring point. Meanwhile, the SPL at the monitoring point before the 6th monitoring point increases, after which the SPL at almost all monitoring points decreases along the train running direction, with an increasingly lower decrease in degree. This can be explained by the fact that the fairing itself brings great noise, and the airflow is disturbed by the leading edge of the fairing first, resulting in some noise reduction at the front end of the fairing cavity. After a period of time, the turbulent flow continues to impact the rear part of the pantograph fairing area, together with the vortex shedding of the pantograph rod, which weakens the noise reduction effect of the fairing. Importantly, compared with case 1, the fairing mainly decreases the noise near the pantograph, and the largest SPL reduction is up to 3 dBA at the 8th monitoring point.

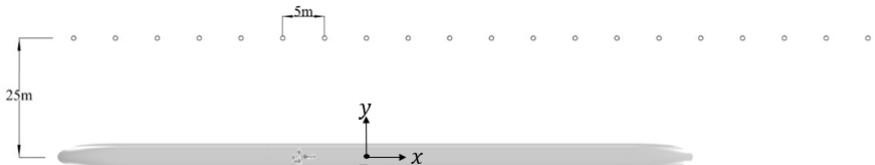

**Figure 11.** Locations of the standard sound monitoring points along the x-direction.

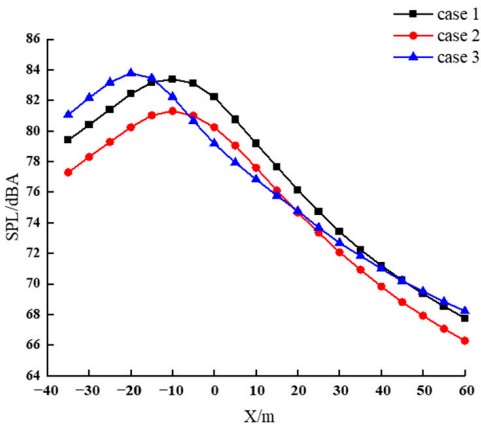

**Figure 12.** Sound pressure level of the monitoring point along the x-direction.

Figure 13 illustrates the 1/3 octave band frequency curves at the maximum SPL monitoring points of the abovementioned three cases. As can be seen in Figure 13, the distribution of aerodynamic noise energy in all the cases is concentrated in 630–2500 Hz, and the SPL gradually decays toward the high frequency and low frequency regions. Furthermore, the noise reduction effect of the sinking platform in the frequency band of 200–1000 Hz is not clear, while the magnitude above 1250 Hz is almost less than case 1 by 2 dBA, which is consistent with the behavior in Figure 12. With regard to case 3, the presence of the fairing greatly increases the noise in the frequency band below 500 Hz, and decreases the noise in the frequency band above 1000 Hz. Consequently, a large SPL maximum is obtained for case 3 in Figure 12.

To further investigate the causes of the difference in SPL energy between the three cases, the spectral characteristics of three critical components of the pantograph, namely, the panhead, the lower arm rod, and the base frame, are analyzed in this section. Figure 14 shows the noise spectrum curves at the maximum SPL monitoring point of each case when the panhead, the lower arm rod, and the base frame are considered as noise sources, respectively. Figure 14a illustrates that when the pantograph is placed on the train roof alone, the main-order and the second-order frequency of 1767 and 3468 Hz occur in the

panhead, with the noise spectrum corresponding to SPL of 69.75 and 52.09 dB, respectively. When the pantograph diversion strategies are ameliorated, the main-order and second-order frequency disappears. In Figure 14b,c, it can be observed that the aerodynamic noise spectrum of the lower arm rod and the base frame in all three cases exhibits a broad frequency characteristic, and there is no clear primary frequency. Moreover, after the installation of the pantograph fairing, the SPL of the lower arm rod and the base frame with energy less than 300 Hz is mainly raised, while the SPL with energy above 1000 Hz is reduced. Clearly, the noise reduction effect of the fairing of the pantograph on the base frame is more salient, which may be due to the fact that the fairing shields part of the incoming flow in the base frame. However, the sinking platform does not have a strong effect on the noise spectrum characteristics of the lower arm rod and the base frame.

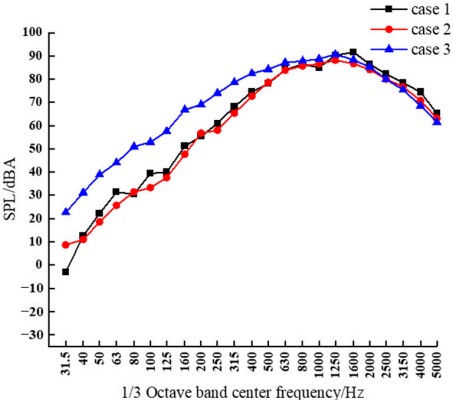

**Figure 13.** The 1/3 octave band frequency spectra of aerodynamic noise at the standard observer points of maximum sound pressure level (case 1: 6th monitoring point; case 2: 6th monitoring point; case 3: 4th monitoring point).

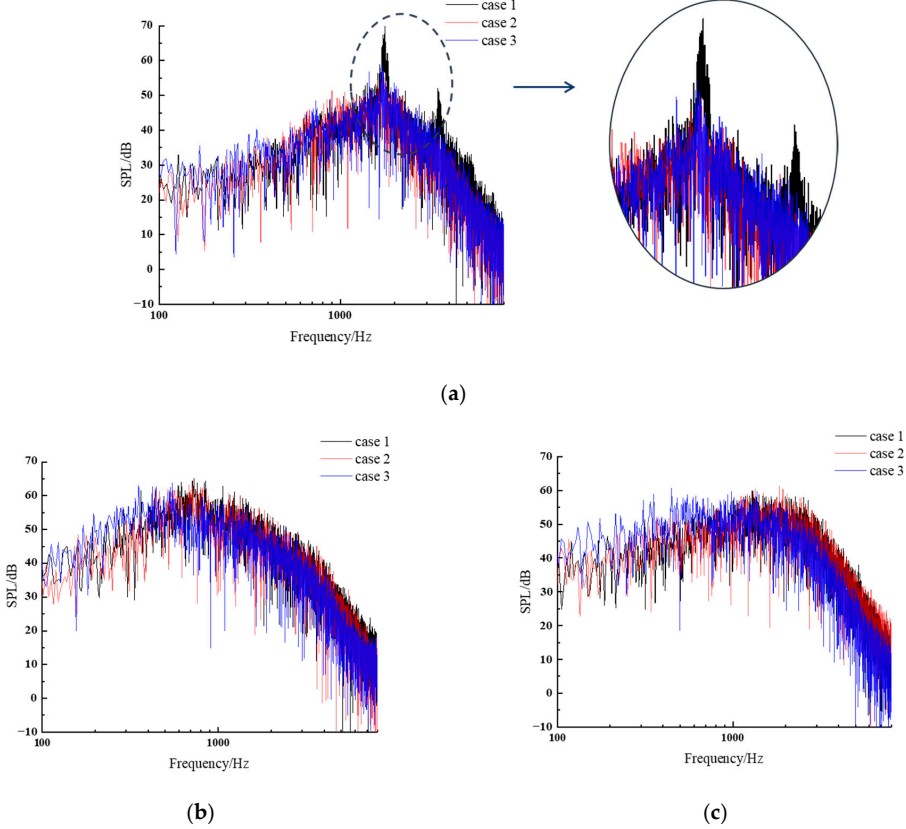

**Figure 14.** Noise spectra of different parts. (**a**) Panhead; (**b**) lower arm rod; (**c**) base frame.

Next, the characteristics of aerodynamic noise at a wide range of frequencies are analyzed. Figure 15 shows the sound source intensity distribution of different cases at 40, 1600, and 8000 Hz, respectively. As can be seen in Figure 15, when there is no pantograph fairing, the sound source intensity in the pantograph area is more uniformly distributed at high frequencies, and the difference in distribution is larger at low and medium frequencies. Meanwhile, the sound source intensity is significantly greater at a frequency of 1600 Hz. In addition, at a low frequency, the high sound source intensity mainly occurs around the camera for case 1, while it appears on the rear surface of the fairing for case 3. A similar distribution can be observed at a medium frequency, although the base frame contributes significantly greatly. Overall, the installation of the pantograph fairing remarkably changed the distribution of the sound source intensity at low and medium frequencies, but had less effect on the distribution at high frequencies.

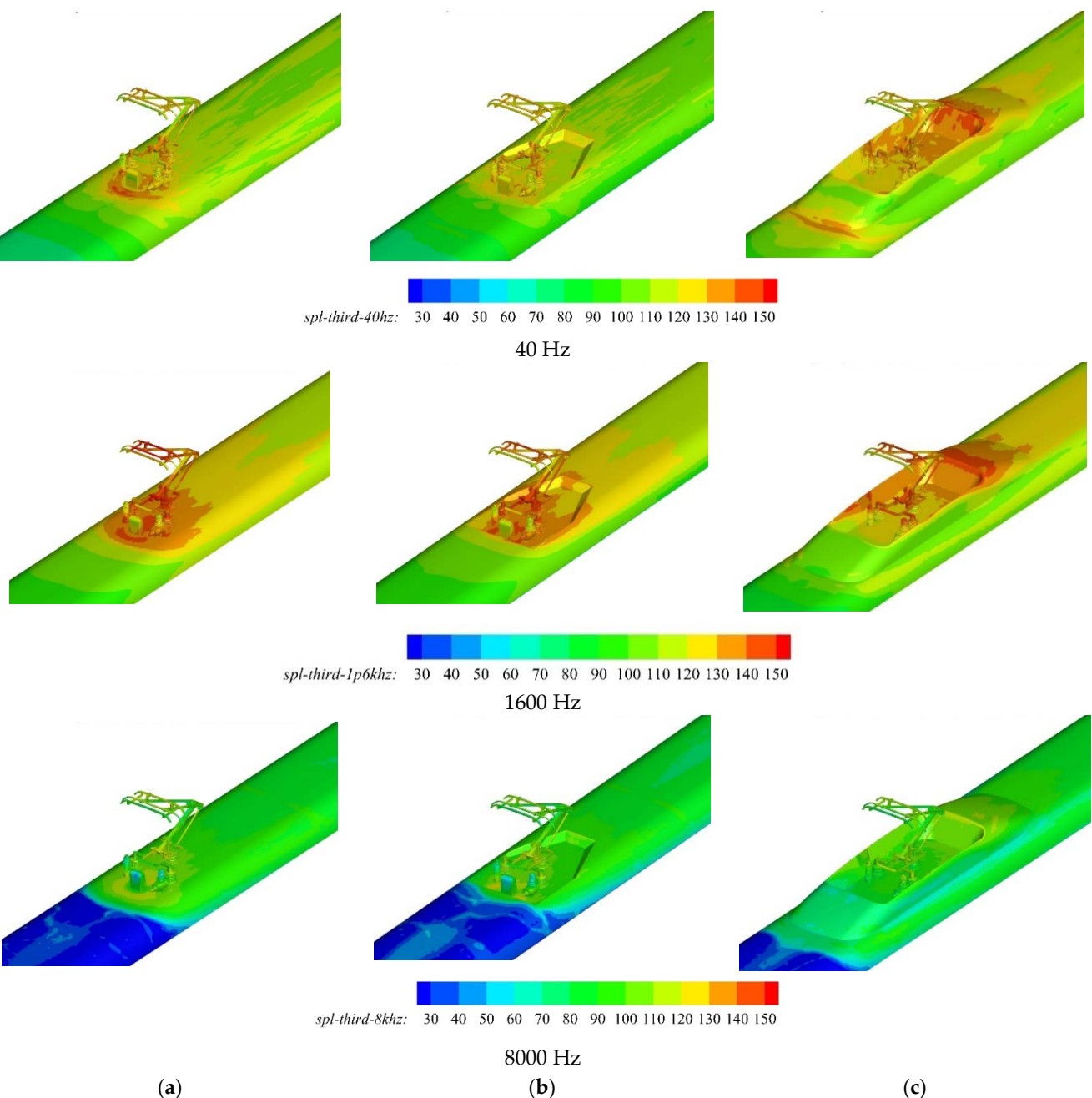

**Figure 15.** Sound source intensity at 40, 1600, and 8000 Hz. (**a**) Case 1; (**b**) case 2; (**c**) case 3.

### 4.2.3. Aerodynamic Noise Directivity

To reveal the difference in aerodynamic noise directivity for different configurations, the pantograph is considered as the sound source and sound monitoring points arranged every 10° on a circle with a radius of 7.5 m are used. As a result, 36 monitoring points ($0° \leq θ \leq 360°$) are set in X-Y plane, and 19 monitoring points ($0° \leq θ \leq 180°$) are set in X-Z and Y-Z planes, respectively. The overall sound pressure level (OASPL) in the three planes is calculated and compared in Figure 16. It can be observed that in the X-Y plane, the noise directivity of cases 1 and 3 is similar, and distinct variations that the OASPL first decreases and then increases appear between $50° \leq θ \leq 90°$, which might be caused by the asymmetry of insulator 1. When the pantograph is installed on the sinking platform, the noise directivity is different from the other two cases, and its variation range is within $60° \leq θ \leq 90°$. In the X-Z plane, as shown in Figure 16b, the OASPL within $90° \leq θ \leq 180°$ is higher than within $0° \leq θ \leq 90°$ for all cases, probably due to the closer proximity to the insulators in this region. In the Y-Z plane, the noise directivity of the three cases is almost symmetrical, and the higher OASPL is mainly distributed above the pantograph, which is $80° \leq θ \leq 100°$.

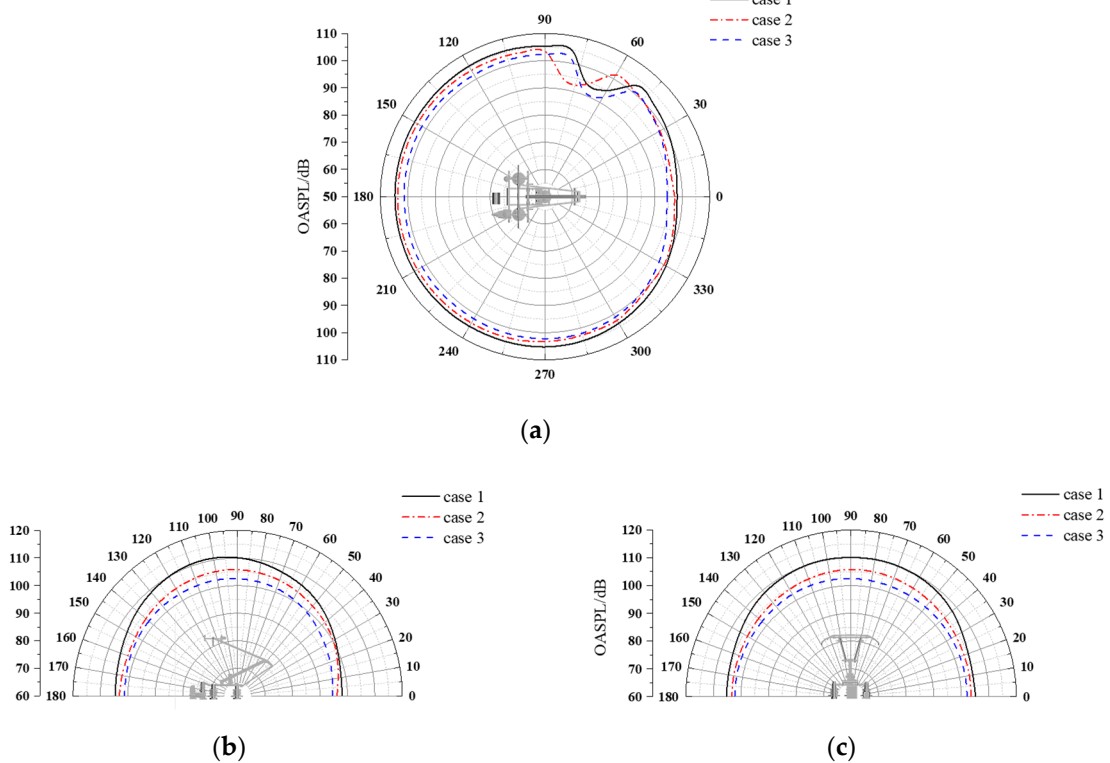

**Figure 16.** Aerodynamic noise directivity caused by pantograph in different planes. (**a**) X-Y plane; (**b**) X-Z plane; (**c**) Y-Z plane.

Of note, in the planes of Y-Z and X-Z, for all monitoring points, the highest OASPL is found for the pantograph alone on the roof, followed by the configuration with the sinking platform on the roof, and the lowest OASPL is found for the pantograph with the fairing. This performance signifies that the fairing effectively reduces the noise induced by the pantograph itself. Meanwhile, the pantograph fairing could play a certain sound barrier role, as well. However, as mentioned above, the noise caused by the fairing itself makes a great contribution and should be considered in a practical scene.

## 5. Conclusions

In this study, large eddy simulation and Ffowcs Williams-Hawkings analogy are used to study the aerodynamic drag and noise characteristics of the pantograph area of a train

roof model with three cars at a speed of 400 km/h with different diversion strategies. Specifically, the flow field characteristics and pressure fluctuation, noise spectrum characteristics, and noise directivity of the computational models are analyzed. The main conclusions are as follows:

(1) The utilization of pantograph fairing on the train roof can ameliorate the aerodynamic drag of HSTs effectively, with a drag reduction rate up to 5.42%. The pantograph noise sources are mainly at the panhead, lower arm rod, and base frame. In addition, the installation of the fairing changes the position of pressure fluctuation when the airflow hits the pantograph.

(2) The performance of aerodynamic noise reduction by the sinking platform and the pantograph fairing are different. The configuration of the pantographs installed on the sinking platform brings about 2 dBA reduction in SPL at 20 sound monitoring points along the train's running direction. In addition, the fairing mainly decreases the noise near the pantograph, and the largest SPL reduction is up to 3 dBA among the monitoring points.

(3) Analysis of 1/3 octave band frequency spectra shows that the noise reduction effect of the sinking platform on the pantograph in the frequency band of 200–1000 Hz is not clear. The installation of the fairing prominently decreases the noise in the frequency band above 1000 Hz. Further analysis of spectral characteristics also shows that the influence of different diversion strategies on the spectral characteristics actually changes the noise spectral characteristics of the panhead.

(4) The installation of the fairing weakens the OASPL of the pantograph effectively. In the X-Y plane, the noise directivity of the pantograph installed alone on the train roof is similar to the fairing installed, while both the noise directivity and OASPL changes when the pantograph is installed on the sinking platform, especially within $50° \leq \theta \leq 90°$; in the X-Z plane, the OASPL for the fairing and sinking platform cases is significantly smaller within $40° \leq \theta \leq 180°$. A similar distribution can be observed in the Y-Z plane.

Overall, the prominent aerodynamic drag reduction effect of the pantograph fairing makes it a competitive solution for HSTs engineering. For the installation of the fairing, it is important to reduce the aerodynamic drag and noise of the faring itself. Furthermore, the combination type of the fairing and pantograph rod deserves further investigation.

**Author Contributions:** Conceptualization, Y.Z. and S.Z. (Siqi Zhou); methodology, H.L., S.Z. (Shishang Zhang), and Z.L.; software, S.Z. (Shishang Zhang) and R.C.; validation, H.L., S.Z. (Siqi Zhou), and R.C.; formal analysis, H.L. and S.Z. (Siqi Zhou); investigation, R.C. and Z.L.; resources, H.L. and S.Z. (Shishang Zhang); data curation, R.C. and Z.L.; writing—original draft preparation, S.Z. (Siqi Zhou); writing—review and editing, H.L., Y.Z., S.Z. (Siqi Zhou), and R.C.; visualization, S.Z. (Siqi Zhou); supervision, R.C. and Z.L.; project administration, H.L. and Z.L.; funding acquisition, H.L. and Y.Z. All authors have read and agreed to the published version of the manuscript.

**Funding:** This research received external funding by the Science and Technology Research and Development Plan of China National Railway Group Co., Ltd. [grant numbers P2020J025, P2021J003, and P2021J036].

**Institutional Review Board Statement:** Not applicable.

**Informed Consent Statement:** Not applicable.

**Data Availability Statement:** Not applicable.

**Acknowledgments:** The authors acknowledge the computing resources provided by the High Performance Computing Public Platform of Central South University, China. This work was supported by the Youth Program of the National Natural Science Foundation of China [grant numbers 11902367 and 12202506], the Youth Program of Natural Science Foundation of Hunan Province, China [grant numbers S2021JJQNJJ2519 and S2021JJQNJJ2716].

**Conflicts of Interest:** The authors declare no conflict of interest.

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
