# Peer review of "Numerical Study on the Aeroacoustic Performance of Different Diversion Strategies in the Pantograph Area of High-Speed Trains at 400 km/h"

_applsci, doi:10.3390/app122110702_

Round 1

Reviewer 1 Report

1.      Regarding the CFD simulation the authors provide some information about the pantograph (only its height) without giving any information about the simulation of the train itself.  For example, in Fig. 10 the whole train is shown. Apparently if the train has been taken into account the used computational space should be further expanded. Namely, instead of using the pantograph’s height, the train dimensions should be used to determine the computational space.

2.      No details are given for the used fairing.

3.      No explanation is given about the used coordinate system x,y,z.

4.      What is the noise in dB without and with the fairing?

5.      It is mentioned that the drag was reduced 5%. Is this for the pantograph alone?

6.      Explain what part of the frequency spectrum has to be supressed to alleviate the noise problem.  

Reviewer 2 Report

This paper presented the numerical investigation of the aerodynamic and aeroacoustic performances of a pantograph installed under different conditions at the speed of 400 km/h. The numerical methodology was validated with the simplified cylinder, and the same method was applied to the realistic pantograph models. The far-field sound amplitudes were predicted by the FW-H method with the surface pressure fluctuations on the pantograph, and the noise sources of each installation condition were numerically explored. The paper is clear and concise except for some details of method and results, and the reviewer recommends the publication in Applied Sciences after clearing the following points.

Line 24: Specify how much the fairing decreased the noise in dB(A) in the abstract.

Page 3-4: A period is missing for some sentences with the equation, like the end of Eq. (1), and Eq. (4).

Line 191: How many grid cells were used in total for this validation case? In addition, please write the size of the grid cells in a spanwise direction, so that readers can easily know the aspect ratio of the boundary mesh.

Line 199: How the authors made the velocity profile of the inlet with the turbulent intensity of 0.8 %? Just the random perturbation or any other model to form the spontaneous turbulence?

Line 244: “… on the train roof. As shown in Figure 6 …” should be “… on the train roof as shown in Figure 6 …”

Figure 6: Please indicate the height of H in the figure (Is it from the bottom of the train to the top of the pantograph?). When there is the sinking platform, the height of the pantograph was increased, and the maximum height was the same for all cases?

Line 260: Please specify y+ for this grid cells of the boundary layer. Is it kept the similar range of the validation case? If not, please describe why you changed that.

Line 267: Why the authors used the pressure inlet and outlet conditions? Did the authors confirm that the inflow velocity was kept at the same value of 400 km/h for all cases? Please describe why the authors did not use the same velocity inlet of the validation case.

Line 271: What was the CFL number? Is it kept small enough for computational accuracy?

Page 9-10: “magnitude velocity” should be “velocity magnitude”

Line 329: Please write Eq. (8) and (9) just before or after mentioning them in the sentence.

Line 348: “Frequency spectra characteristics of SPL” should be “Spectral characteristics of SPL”

Figure10: Please indicate the axis directions and the origin of each coordinate in the figure.

Figure 11 and 12: In Fig. 11, the maximum amplitudes of case 1 and 3 were almost the same at 84 dB(A), and the maximum of case 2 decreased to 81 dB(A). However, in Fig. 12, the overall amplitudes of case 3 were the maximum in three cases for the most of frequency range. Is it correct? The figure description says that the frequency spectra of maximum SPL monitoring points were plotted for three cases. If the authors plotted the maximum for each case in Fig. 11, the spectra of Fig. 12 should be case3 ≈ case 1 >> case 2.

Figure 13 (a): Since the spectra of cases 2 and 3 were overlapped by that of case 1, readers cannot see whether the peaks of case 1 disappeared in the other case or not.    

Reviewer 3 Report

In this Study, the researchers focused on the effect of pantograph shape and cavity on the aerodynamic performance and aero noise of a high speed train. A great LES work is conducted by the authors and the performed analysis is good enough for a scientific paper. There are some minor problems and questions which are presented in the following:

1.  First paragraph of the introduction is not directly related to the references number 1 and 2, either the references or the paragraph should be changed.

2.  The validation of the results is conducted for a 2D case; however, the research is performed for a 3D geometry. Therefore, a major concern is raised about the validation part of this study.

3. Based on Figure 5, the domain of the research is not selected appropriately. According to the validation reference, the selected domain length is 20 times bigger than the cylinder diameter; nevertheless, within the main study this parameter is selected 2 times of the train length.

4.  Why the frequency of 40 Hz, 1600 Hz, and 8000 Hz are selected for the Figure 14 and focused on the SPL distribution?
